# Regularization With Stochastic Transformations and Perturbations for Deep Semi-Supervised Learning

**Mehdi Sajjadi**        **Mehran Javanmardi**        **Tolga Tasdizen**

Department of Electrical and Computer Engineering
University of Utah
{mehdi, mehran, tolga}@sci.utah.edu

## Abstract

Effective convolutional neural networks are trained on large sets of labeled data. However, creating large labeled datasets is a very costly and time-consuming task. Semi-supervised learning uses unlabeled data to train a model with higher accuracy when there is a limited set of labeled data available. In this paper, we consider the problem of semi-supervised learning with convolutional neural networks. Techniques such as randomized data augmentation, dropout and random max-pooling provide better generalization and stability for classifiers that are trained using gradient descent. Multiple passes of an individual sample through the network might lead to different predictions due to the non-deterministic behavior of these techniques. We propose an unsupervised loss function that takes advantage of the stochastic nature of these methods and minimizes the difference between the predictions of multiple passes of a training sample through the network. We evaluate the proposed method on several benchmark datasets.

## 1 Introduction

Convolutional neural networks (ConvNets) [1, 2] achieve state-of-the-art accuracy on a variety of computer vision tasks, including classification, object localization, detection, recognition and scene labeling [3, 4]. The advantage of ConvNets partially originates from their complexity (large number of parameters), but this can result in overfitting without a large amount of training data. However, creating a large labeled dataset is very costly. A notable example is the 'ImageNet' [5] dataset with 1000 category and more than 1 million training images. The state-of-the-art accuracy of this dataset is improved every year using ConvNet-based methods (e.g., [6, 7]). This dataset is the result of significant manual effort. However, with around 1000 images per category, it barely contains enough training samples to prevent the ConvNet from overfitting [7]. On the other hand, unlabeled data is cheap to collect. For example, there are numerous online resources for images and video sequences of different types. Therefore, there has been an increased interest in exploiting the readily available unlabeled data to improve the performance of ConvNets.

Randomization plays an important role in the majority of learning systems. Stochastic gradient descent, dropout [8], randomized data transformation and augmentation [9] and many other training techniques that are essential for fast convergence and effective generalization of the learning functions introduce some non-deterministic behavior to the learning system. Due to these uncertainties, passing a single data sample through a learning system multiple times might lead to different predictions. Based on this observation, we introduce an unsupervised loss function optimized by gradient descent that takes advantage of this randomization effect and minimizes the difference in predictions of multiple passes of a data sample through the network during the training phase, which leads to better generalization in testing time. The proposed unsupervised loss function specifically regularizes the network based on the variations caused by randomized data augmentation, dropout and randomized max-pooling schemes. This loss function can be combined with any supervised loss function. In this

paper, we apply the proposed unsupervised loss function to ConvNets as a state-of-the-art supervised classifier. We show through numerous experiments that this combination leads to a competitive semi-supervised learning method.

## 2 Related Work

There are many approaches to semi-supervised learning in general. Self-training and co-training [10, 11] are two well-known classic examples. Another set of approaches is based on generative models, for example, methods based on Gaussian Mixture Models (GMM) and Hidden Markov Models (HMM) [12]. These generative models generally try to use unlabeled data in modeling the joint probability distribution of the training data and labels. Transductive SVM (TSVM) [13] and S3VM [14] are another semi-supervised learning approach that tries to find a decision boundary with a maximum margin on both labeled and unlabeled data. A large group of semi-supervised methods is based on graphs and the similarities between the samples [15, 16]. For example, if a labeled sample is similar to an unlabeled sample, its label is assigned to that unlabeled sample. In these methods, the similarities are encoded in the edges of a graph. Label propagation [17] is an example of these methods in which the goal is to minimize the difference between model predictions of two samples with large weighted edge. In other words, similar samples tend to get similar predictions.

In this paper, our focus is on semi-supervised deep learning. There has always been interest in exploiting unlabeled data to improve the performance of ConvNets. One approach is to use unlabeled data to pre-train the filters of ConvNet [18, 19]. The goal is to reduce the number of training epochs required to converge and improve the accuracy compared to a model trained by random initialization. Predictive sparse decomposition (PSD) [20] is one example of these methods used for learning the weights in the filter bank layer. The works presented in [21] and [22] are two recent examples of learning features by pre-training ConvNets using unlabeled data. In these approaches, an auxiliary target is defined for a pair of unlabeled images [21] or a pair of patches from a single unlabeled image [22]. Then a pair of ConvNets is trained to learn descriptive features from unlabeled images. These features can be fine-tuned for a specific task with a limited set of labeled data. However, many recent ConvNet models with state-of-the-art accuracy start from randomly initialized weights using techniques such as Xavier's method [23, 6]. Therefore, approaches that make better use of unlabeled data during training instead of just pre-training are more desired.

Another example of semi-supervised learning with ConvNets is region embedding [24], which is used for text categorization. The work in [25] is also a deep semi-supervised learning method based on embedding techniques. Unlabeled video frames are also being used to train ConvNets [26, 27]. The target of the ConvNet is calculated based on the correlations between video frames. Another notable example is semi-supervised learning with ladder networks [28] in which the sums of supervised and unsupervised loss functions are simultaneously minimized by backpropagation. In this method, a feedforward model, is assumed to be an encoder. The proposed network consists of a noisy encoder path and a clean one. A decoder is added to each layer of the noisy path. This decoder is supposed to reconstruct a clean activation of each layer. The unsupervised loss function is the difference between the output of each layer in clean path and its corresponding reconstruction from the noisy path.

Another approach by [29] is to take a random unlabeled sample and generate multiple instances by randomly transforming that sample multiple times. The resulting set of images forms a surrogate class. Multiple surrogate classes are produced and a ConvNet is trained on them. One disadvantage of this method is that it does not scale well with the number of unlabeled examples because a separate class is needed for every training sample during unsupervised training. In [30], the authors propose a mutual-exclusivity loss function that forces the set of predictions for a multiclass dataset to be mutually-exclusive. In other words, it forces the classifier's prediction to be close to one only for one class and zero for the others. It is shown that this loss function makes use of unlabeled data and pushes the decision boundary to a less dense area of decision space.

Another set of works related to our approach try to restrict the variations of the prediction function. Tangent distance and tangent propagation proposed by [31] enforce local classification invariance with respect to the transformations of input images. Here, we propose a simpler method that additionally minimizes the internal variations of the network caused by dropout and randomized pooling and leads to state-of-the-art results on MNIST (with 100 labeled samples), CIFAR10 and CIFAR100. Another example is Slow Feature Analysis (SFA) (e.g., [32] and [33]) that encourages the representations of temporally close data to exhibit small differences.

## 3 Method

Given any training sample, a model's prediction should be the same under any random transformation of the data and perturbations to the model. The transformations can be any linear and non-linear data augmentation being used to extend the training data. The disturbances include dropout techniques and randomized pooling schemes. In each pass, each sample can be randomly transformed or the hidden nodes can be randomly activated. As a result, the network's prediction can be different for multiple passes of the same training sample. However, we know that each sample is assigned to only one class. Therefore, the network's prediction is expected to be the same despite transformations and disturbances. We introduce an unsupervised loss function that minimizes the mean squared differences between different passes of an individual training sample through the network. Note that we do not need to know the label of a training sample in order to enforce this loss. Therefore, the proposed loss function is completely unsupervised and can be used along with supervised training as a semi-supervised learning method. Even if we don't have a separate unlabeled set, we can apply the proposed loss function on samples of labeled set to enforce stability.

Here, we formally define the proposed unsupervised loss function. We start with a dataset with $N$ training samples and $C$ classes. Let us assume that $\mathbf{f}^j(\mathbf{x}_i)$ is the classifier's prediction vector on the $i$'th training sample during the $j$'th pass through the network. We assume that each training sample is passed $n$ times through the network. We define the $T^j(\mathbf{x}_i)$ to be a random linear or non-linear transformation on the training sample $\mathbf{x}_i$ before the $j$'th pass through the network. The proposed loss function for each data sample is:

$$l_{\mathcal{U}}^{\text{TS}} = \sum_{i=1}^{N} \sum_{j=1}^{n-1} \sum_{k=j+1}^{n} \|\mathbf{f}^j(T^j(\mathbf{x}_i)) - \mathbf{f}^k(T^k(\mathbf{x}_i))\|_2^2 \tag{1}$$

Where 'TS' stands for transformation/stability. We pass a training sample through the network $n$ times. In each pass, the transformation $T^j(\mathbf{x}_i)$ produces a different input to the network from the original training sample. In addition, each time the randomness inside the network, which can be caused by dropout or randomized pooling schemes, leads to a different prediction output. We minimize the sum of squared differences between each possible pair of predictions. We can minimize this objective function using gradient descent. Although Eq. 1 is quadratically dependent on the number of augmented versions of the data ($n$), calculation of loss and gradient is only based on the prediction vectors. So, the computing cost is negligible even for large $n$. Note that recent neural-network-based methods are optimized on batches of training samples instead of a single sample (batch vs. online training). We can design batches to contain replications of training samples so we can easily optimize this transformation/stability loss function. If we use data augmentation, we put different transformed versions of an unlabeled data in the mini-batch instead of replication. This unsupervised loss function can be used with any backpropagation-based algorithm. Even though, every mini-batch contains replications of a training sample, these are used to calculate a single backpropagation signal avoiding gradient bias and not adversely affecting convergence. It is also possible to combine this loss with any supervised loss function. We reserve part of the mini-batch for labeled data which are not replicated.

As mentioned in section 2, mutual-exclusivity loss function of [30] forces the classifier's prediction vector to have only one non-zero element. This loss function naturally complements the transformation/stability loss function. In supervised learning, each element of the prediction vector is pushed towards zero or one depending on the corresponding element in label vector. The proposed loss minimizes the $l_2$-norm of the difference between predictions of multiple transformed versions of a sample, but it does not impose any restrictions on the individual elements of a single prediction vector. As a result, each prediction vector might be a trivial solution instead of a valid prediction due to lack of labels. Mutual-exclusivity loss function forces each prediction vector to be valid and prevents trivial solutions. This loss function for the training sample $\mathbf{x}_i$ is defined as follows:

$$l_{\mathcal{U}}^{\text{ME}} = \sum_{i=1}^{N} \sum_{j=1}^{n} \left( -\sum_{k=1}^{C} f_k^j(\mathbf{x}_i) \prod_{l=1, l \neq k}^{C} (1 - f_l^j(\mathbf{x}_i)) \right) \tag{2}$$

Where 'ME' stands for mutual-exclusivity. $f_k^j(\mathbf{x}_i)$ is the $k$-th element of prediction vector $\mathbf{f}^j(\mathbf{x}_i)$. In the experiments, we show that the combination of both loss functions leads to further improvements in the accuracy of the models. We define the combination of both loss functions as transforma-

tion/stability plus mutual-exclusivity loss function:

$$l_{\mathcal{U}} = \lambda_1 l_{\mathcal{U}}^{\text{ME}} + \lambda_2 l_{\mathcal{U}}^{\text{TS}} \tag{3}$$

## 4 Experiments

We show the effect of the proposed unsupervised loss functions using ConvNets on MNIST [2], CIFAR10 and CIFAR100 [34], SVHN [35], NORB [36] and ILSVRC 2012 challenge [5]. We use two frameworks to implement and evaluate the proposed loss function. The first one is cuda-convnet [37], which is the original implementation of the well-known AlexNet model. The second framework is the sparse convolutional networks [38] with fractional max-pooling [39], which is a more recent implementation of ConvNets achieving state-of-the-art accuracy on CIFAR10 and CIFAR100 datasets. We show through different experiments that by using the proposed loss function, we can improve the accuracy of the models trained on a few labeled samples on both implementations. In Eq. 1, we set $n$ to be 4 for experiments conducted using cuda-convnet and 5 for experiments performed using sparse convolutional networks. Sparse convolutional network allows for any arbitrary batch sizes. As a result, we tried different options for $n$ and $n = 5$ is the optimal choice. However, cuda-convnet allows for mini-batches of size 128. Therefore, it is not possible to use $n = 5$. Instead, we decided to use $n = 4$. In practice the difference is insignificant. We used MNIST to find the optimal $n$. We tried different $n$ up to 10 and did not observe improvements for $n$ larger than 5. It must be noted that replicating a training sample four or five times does not necessarily increase the computational complexity with the same factor. Based on the experiments, with higher $n$ fewer training epochs are required for the models to converge. We perform multiple experiments for each dataset. We use the available training data of each dataset to create two sets: labeled and unlabeled. We do not use the labels of the unlabeled set during training. It must be noted that for the experiments with data augmentation, we apply data augmentation to both labeled and unlabeled set. We compare models that are trained only on the labeled set with models that are trained on both the labeled set and the unlabeled set using the unsupervised loss function. We show that by using the unsupervised loss function, we can improve the accuracy of classifiers on benchmark datasets. For experiments performed using sparse convolutional network, we describe the network parameters using the format adopted from the original paper [39]:

$$(10kC2 - FMP\sqrt{2})_5 - C2 - C1$$

In the above example network, $10k$ is the number of maps in the $k$'th convolutional layer. In this example, $k = 1, 2, ..., 5$. $C2$ specifies that convolutions use a kernel size of 2. $FMP\sqrt{2}$ indicates that convolutional layers are followed by a fractional max-pooling (FMP) layer [39] that reduces the size of feature maps by a factor of $\sqrt{2}$. As mentioned earlier, the mutual-exclusivity loss function of [30] complements the transformation/stability loss function. We implement that loss function in both cuda-convnet and sparse convolutional networks as well. We experimentally choose $\lambda_1$ and $\lambda_2$ in Eq. 3. However, the performance of the models is not overly sensitive to these parameters, and in most of the experiments it is fixed to $\lambda_1 = 0.1$ and $\lambda_2 = 1$.

### 4.1 MNIST

MNIST is the most frequently used dataset in the area of digit classification. It contains 60000 training and 10000 test samples of size $28 \times 28$ pixels. We perform experiments on MNIST using a sparse convolutional network with the following architecture: $(32kC2 - FMP\sqrt{2})_6 - C2 - C1$. We use dropout to regularize the network. The ratio of dropout gradually increases from the first layer to the last layer. We do not use any data augmentation for this task. In other words, $T^j(\mathbf{x}_i)$ of Eq. 1 is identity function for this dataset. In this case, we take advantage of the random effects of dropout and fractional max-pooling using the unsupervised loss function. We randomly select 10 samples from each class (total of 100 labeled samples). We use all available training data as the unlabeled set. First, we train a model based on this labeled set only. Then, we train models by adding unsupervised loss functions. In separate experiments, we add transformation/stability loss function, mutual-exclusivity loss function and the combination of both. Each experiment is repeated five times with a different random subset of training samples. We repeat the same set of experiments using 100% of MNIST training samples. The results are given in Table 1. We can see that the proposed loss significantly improves the accuracy on test data. We also compare the results with ladder networks [28]. Combination of both loss functions reduces the error rate to 0.55%

$\pm$ 0.16 which is the state-of-the-art for the task of MNIST with 100 labeled samples to the best of our knowledge. The state-of-the-art error rate on MNIST using all training data without data augmentation is 0.24% [40]. It can be seen that we can achieve a close accuracy by using only 100 labeled samples.

Table 1: Error rates (%) on test set for MNIST (mean % $\pm$ std).

|  | labeled data only | transform /stability loss | mut-excl loss [30] | both losses | ladder net. [28] | ladder net baseline [28] |
|---|---|---|---|---|---|---|
| 100 : | $5.44 \pm 1.48$ | $0.76 \pm 0.61$ | $3.92 \pm 1.12$ | $\mathbf{0.55 \pm 0.16}$ | $0.89 \pm 0.50$ | $6.43 \pm 0.84$ |
| all: | $0.32 \pm 0.02$ | $0.29 \pm 0.02$ | $0.30 \pm 0.03$ | $\mathbf{0.27 \pm 0.02}$ | - | 0.36 |

## 4.2   SVHN and NORB

SVHN is another digit classification task similar to MNIST. This dataset contains about 70000 images for training and more than 500000 easier images [35] for validation. We do not use the validation set. The test set contains 26032 images, which are RGB images of size $32 \times 32$. Generally, SVHN is a more difficult task compared to MNIST because of the large variations in the images. We do not perform any pre-processing for this dataset. We simply convert the color images to grayscale by removing hue and saturation information. NORB is a collection of stereo images in six classes. The training set contains 10 folds of 29160 images. It is common practice to use only the first two folds for training. The test set contains two folds, totaling 58320. The original images are $108 \times 108$. However, we scale them down to $48 \times 48$ similar to [9]. We perform experiments on these two datasets using both cuda-convnet and sparse convolutional network implementations of the unsupervised loss function.

In the first set of experiments, we use cuda-convnet to train models with different ratios of labeled and unlabeled data. We randomly choose 1%, 5%, 10%, 20% and 100% of training samples as labeled data. All of the training samples are used as the unlabeled set. For each labeled set, we train four models using cuda-convnet. The first model uses labeled set only. The second model is trained on unlabeled set using mutual-exclusivity loss function in addition to the labeled set. The third model is trained on the unlabeled set using the transformation/stability loss function in addition to the labeled set. The last model is also trained on both sets but combines two unsupervised loss functions. Each experiment is repeated five times. For each repetition, we use a different subset of training samples as labeled data. The cuda-convnet model consists of two convolutional layers with 64 maps and kernel size of 5, two locally connected layers with 32 maps and kernel size 3. Each convolutional layer is followed by a max-pooling layer. A fully connected layer with 256 nodes is added before the last layer. We use data augmentation for these experiments. $T^j(\mathbf{x}_i)$ of Eq. 1 crops every training sample to $28 \times 28$ for SVHN and $44 \times 44$ for NORB at random locations. $T^j(\mathbf{x}_i)$ also randomly rotates training samples up to $\pm 20°$. These transformations are applied to both labeled and unlabeled sets. The results are shown in Figure 1 for SVHN and Figure 2 for NORB. Each point in the graph is the mean error rate of five repetitions. The error bars show the standard deviation of these five repetitions. As expected, we can see that in all experiments the classification accuracy is improved as we add more labeled data. However, we observe that for each set of labeled data we can improve the results by using the proposed unsupervised loss functions. We can also see that when the number of labeled samples is small, the improvement is more significant. For example, when we use only 1% of labeled data, we gain an improvement in accuracy of about 2.5 times by using unsupervised loss functions. As we add more labeled samples, the difference in accuracy between semi-supervised and supervised approaches becomes smaller. Note that the combination of transformation/stability loss function and mutual-exclusivity loss function improves the accuracy even further. As mentioned earlier, these two unsupervised loss functions complement each other. Therefore, in most of the experiments we use the combination of two unsupervised loss functions.

We perform another set of experiments on these two datasets using sparse convolutional networks as a state-of-the-art classifier. We create five sets of labeled data. For each set, we randomly pick a different 1% subset of training samples as labeled set and all training data as unlabeled set. We train two models: the first trained only on labeled data, and the second using the labeled set and a combination of both unsupervised losses. Similarly, we train models using all available training data as both the labeled set and unlabeled set. We do not use data augmentation for any of these experiments. In other

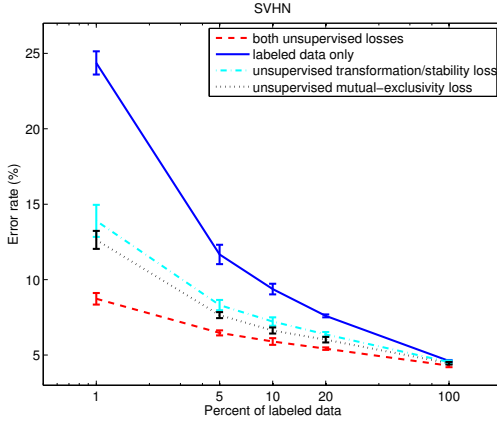

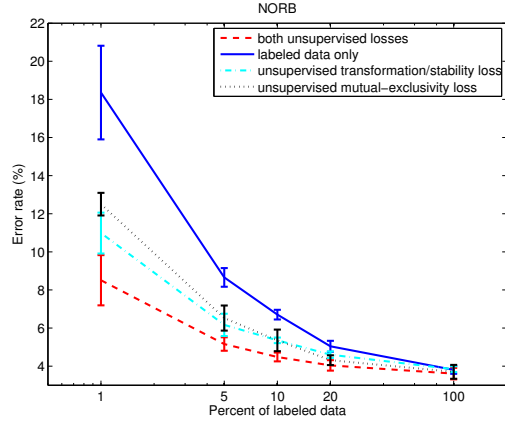

Figure 1: SVHN dataset: semi-supervised learning vs. training with labeled data only.

Figure 2: NORB dataset: semi-supervised learning vs. training with labeled data only.

words, $T^j(\mathbf{x}_i)$ of Eq. 1 is identity function. As a result, dropout and random max-pooling are the only sources of variation in this case. We use the following model: $(32kC2 - FMP\sqrt[3]{2})_{12} - C2 - C1$. Similar to MNIST, we use dropout to regularize the network. Again, the ratio of dropout gradually increases from the first layer to the last layer. The results (average of five error rates) are shown in Table 2. Here, we can see that by using unsupervised loss functions we can significantly improve the accuracy of the classifier by trying to minimize the variation in prediction of the network. In addition, for NORB dataset we can observe that by using only 1% of labeled data and applying unsupervised loss functions, we can achieve accuracy that is close to the case when we use 100% of labeled data.

Table 2: Error on test data for SVHN and NORB with 1% and 100% of data (mean % ± std).

|  | SVHN | | NORB | |
|---|---|---|---|---|
|  | 1% of data | 100% of data | 1% of data | 100% of data |
| labeled data only: | $12.25 \pm 0.80$ | $2.28 \pm 0.05$ | $10.01 \pm 0.81$ | $1.63 \pm 0.12$ |
| semi-supervised: | $6.03 \pm 0.62$ | $2.22 \pm 0.04$ | $2.15 \pm 0.37$ | $1.63 \pm 0.07$ |

## 4.3 CIFAR10

CIFAR10 is a collection of 60000 tiny $32 \times 32$ images of 10 categories (50000 for training and 10000 for test). We use sparse convolutional networks to perform experiments on this dataset. For this dataset, we create 10 labeled sets. Each set contains 4000 samples that are randomly picked from the training set. All 50000 training samples are used as unlabeled set. We train two sets of models on these data. The first set of models is trained on labeled data only, and the other set of models is trained on the unlabeled set using a combination of both unsupervised loss functions in addition to the labeled set. For this dataset, we do not perform separate experiments for two unsupervised loss functions because of time constraints. However, based on the results from MNIST, SVHN and NORB, we deduce that the combination of both unsupervised losses provides improved accuracy. We use data augmentation for these experiments. Similar to [39], we perform affine transformations, including randomized mix of translations, rotations, flipping, stretching and shearing operations by $T^j(\mathbf{x}_i)$ of Eq. 1. Similar to [39], we train the network without transformations for the last 10 epochs. We use the following parameters for the models: $(32kC2 - FMP\sqrt[3]{2})_{12} - C2 - C1$. We use dropout, and its ratio gradually increases from the first layer to the last layer. The results are given in Table 3. We also compare the results to ladder networks [28]. The model in [28] does not use data augmentation. We can see that the combination of unsupervised loss functions on unlabeled data improves the accuracy of the models. In another set of experiments, we use all available training data as both labeled and unlabeled sets. We train a network with the following parameters: $(96kC2 - FMP\sqrt[3]{2})_{12} - C2 - C1$. We use affine transformations for this task too. Here again, we use transformation/stability plus the mutual-exclusivity loss function. We repeat this experiments five times and achieve **3.18% ± 0.1** mean and standard deviation error rate. The

Table 3: Error rates on test data for CIFAR10 with 4000 labeled samples (mean % ± std).

| | transformation/stability+mutual-exclusivity | ladder networks [28] |
|---|---|---|
| labeled data only: | $13.60 \pm 0.24$ | $23.33 \pm 0.61$ |
| semi-supervised: | $11.29 \pm 0.24$ | $20.40 \pm 0.47$ |

state-of-the-art error rate for this dataset is 3.47%, achieved by the fractional max-pooling method [39] but obtained with a larger model ($160n$ vs. $96n$). We perform a single run experiment with $160n$ model and achieve the error rate of **3.00%**. Similar to [39], we perform 100 passes during test time. Here, we surpass state-of-the-art accuracy by adding unsupervised loss functions.

## 4.4 CIFAR100

CIFAR100 is also a collection of 60000 tiny images of size $32 \times 32$. This dataset is similar to CIFAR10. However, it contains images of 100 categories compared to 10. Therefore, we have a smaller number of training samples per category. Similar to CIFAR10, we perform experiments on this dataset using sparse convolutional networks. We use all available training data as both labeled and unlabeled sets. The state-of-the-art error rate for this dataset is 23.82%, obtained by fractional max-pooling [39] on sparse convolutional networks. The following model was used to achieve this error rate: $(96kC2 - FMP\sqrt[3]{2})_{12} - C2 - C1$. Dropout was also used with a ratio increasing from the first layer to the last layer. We use the same model parameters and add transformation/stability plus the mutual-exclusivity loss function. Similar to [39], we do not use data augmentation for this task ($T^j(\mathbf{x}_i)$ of Eq. 1 is identity function). Therefore, the proposed loss function minimizes the randomness effect due to dropout and max-pooling. We achieve **21.43% ± 0.16** mean and standard deviation error rate, which is the state-of-the-art for this task. We perform 12 passes during the test time similar to [39].

## 4.5 ImageNet

We perform experiments on the ILSVRC 2012 challenge. The training data consists of 1281167 natural images of different sizes from 1000 categories. We create five labeled datasets from available training samples. Each dataset consists of 10% of training data. We form each dataset by randomly picking a subset of training samples. All available training data is used as the unlabeled set. We use cuda-convnet to train AlexNet model [7] for this dataset. Similar to [7], all images are re-sized to $256 \times 256$. We also use data augmentation for this task following steps of [7], i.e., $T^j(\mathbf{x}_i)$ of Eq. 1 performs random translations, flipping and color noise. We train two models on each labeled dataset. One model is trained using labeled data only. The other model is trained on both labeled and unlabeled set using the transformation/stability plus mutual-exclusivity loss function. At each iteration, we generate four different transformed versions of each unlabeled sample. So, each unlabeled sample is forward passed through the network four times. Since we use all training data as unlabeled set, the computational cost of each iteration is roughly quadrupled. But, in practice we found that when we use 10% of training data as labeled set, the network converges in 20 epochs instead of standard 90 epochs of AlexNet model. So, overall cost of our method for ImageNet is less than or equal to AlexNet. The results on validation set are shown in Table 4. We also compare the results to the model trained on the mutual-exclusivity loss function only and reported in [30]. We can see that even for a large dataset with many categories, the proposed unsupervised loss function improves the classification accuracy. The error rate of a single AlexNet model on validation set of ILSVRC 2012 using all training data is 18.2% [7].

Table 4: Error rates (%) on validation set for ILSVR 2012 (Top-5).

| | rep 1 | rep 2 | rep 3 | rep 4 | rep 5 | mean ± std | mutual xcl [30] | [21] ~1.5% of data |
|---|---|---|---|---|---|---|---|---|
| labeled only: | 45.73 | 46.15 | 46.06 | 45.57 | 46.08 | $45.91 \pm 0.25$ | 45.63 | 85.9 |
| semi-sup: | 39.50 | 39.99 | 39.94 | 39.70 | 40.08 | $39.84 \pm 0.23$ | 42.90 | 84.2 |

## 5 Discussion

We can see that the proposed loss function can improve the accuracy of a ConvNet regardless of the architecture and implementation. We improve the accuracy of two relatively different implementations of ConvNets, i.e., cuda-convnet and sparse convolutional networks. For SVHN and NORB, we do not use dropout or randomized pooling for the experiments performed using cuda-convnet. Therefore, the only source of variation in different passes of a sample through the network is random transformations (translation and rotation). For the experiments performed using sparse convolutional networks on these two datasets, we do not use data transformation. Instead, we use dropout and randomized pooling. Based on the results, we can see that in both cases we can significantly improve the accuracy when we have a small number of labeled samples. For CIFAR100, we achieve state-of-the-art error rate of 21.43% by taking advantage of the variations caused by dropout and randomized pooling. In ImageNet and CIFAR10 experiments, we use both data transformation and dropout. For CIFAR10, we also have randomized pooling and achieve the state-of-the-art error rate of 3.00%. In MNIST experiments with 100 labeled samples and NORB experiments with 1% of labeled data, we achieve accuracy reasonably close to the case when we use all available training data by applying mutual-exclusivity loss and minimizing the difference in predictions of multiple passes caused by dropout and randomized pooling.

## 6 Conclusion

In this paper, we proposed an unsupervised loss function that minimizes the variations in different passes of a sample through the network caused by non-deterministic transformations and randomized dropout and max-pooling schemes. We evaluated the proposed method using two ConvNet implementations on multiple benchmark datasets. We showed that it is possible to achieve significant improvements in accuracy by using the transformation/stability loss function along with mutual-exclusivity of [30] when we have a small number of labeled data available.

**Acknowledgments**

This work was supported by NSF IIS-1149299.

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
