[Reviews · NeurIPS 2016]

Reviewer 1

Summary

The paper explores the potential of using unlabeled data (transformed versions of the training set) for increasing the generalization properties of convolutional neural network. It formulates an unsupervised loss function, which can be used in conjunction with standard supervised training, for minimizing the variability of multiple passes of the same point through the network due to various stochastic regularization aspects of training (dropout, stochastic pooling etc.) or various augmentation transformations. The loss depends on minimizing the prediction difference among all combinations of transformed or perturbed points (without using label information), along with enforcing mutual exclusivity in the values of the prediction output vector. The large body of this work is of experimental nature, with multiple benchmarks using 2 different network architectures. The unsupervised loss improves performance consistently, especially in the low sample regime, and achieves sate-of-the-art performance in CIFAR10 and CIFAR100.

Qualitative Assessment

This work proposes to use semi-supervised learning, in the form of an unsupervised loss term, for improving the regularization capacity of CNNs. The idea (and the proposed loss) is conceptually simple and enforces stability explicitly by minimizing the difference between predictions corresponding to the same input data point. The paper focuses mainly on the experimental side, devoting the largest part in presenting results when adding the new loss on standard supervised CNNs. This is the stronger aspect of this work, with the weaker being the lack (or the definition) of baselines and the lack of some form of theoretical justification, derivation or discussion. Novelty/originality: The main contribution is the application of the unsupervised loss term for controlling the stability of the predictions under transformations or stochastic variability. The mutual exclusivity term is from previous work. There is merit in the experimental results promoting the case of semi-supervised learning in training of large neural networks. Technical quality: Extensive experimental evaluations are given on well known vision datasets with two different models and two different perturbation types (transforming the input and randomization during training), focusing on comparing performance with and without the new loss. Comparisons with one other relevant method is included (but not reproduced or controlled for). Baselines could be further improved (e.g. by including supervised training on augmented labeled data, using the same transformations like in the unsupervised case). Potential impact: The experimental study makes the case for using unlabeled data to complement supervised training (like in [30]) and enforcing stability, through a simple loss, on the predictions. In addition, results that achieve state-of-the-art are reported on CIFAR10 and CIFAR100. Q (analysis): The proposed loss is not studied theoretically or discussed in the context of the underlying perturbations. As a matter of fact many of these techniques play the role of additional regularization, while data augmentation acts as an expansion of the training set. For the latter, there is obvious purpose in enforcing the mapping to be the same (same model, perturbed input). The former, however, seems like enforcing different models to give the same prediction, thus canceling out the effects of randomization in the process). Some discussion would be useful here (e.g.see the equivalence of dropout to model averaging). Q (efficiency): How does the proposed loss scale up with the number of transformations in the augmented set. Since (from (1)), there is a quadratic dependency on the number of augmented versions of the data n, which values of n are practical for computing the loss and being useful for the type of transformations used? The authors use n = 4, 5 here. These values may be small for the case augmentation via geometric transformations (e.g. affine). An empirical study on the role of n (and the trade-off between performance and efficiency) might be insightful. Q (implementation): How is the loss computed during training? The paper mentions creating min-batches with replicates (or transformations) of the same data points. Will this create gradient bias or affect convergence? Q (baselines): See below on experiments. Q: The authors make a comment (Sec. 3, line 102-103) that the loss could be potentially used on labeled samples also to enforce stability. This could potentially be another interesting baseline to explore (in connection to unsupervised vs. supervised training). Clarity and presentation: - Paragraph 1 in Sec. 2 (Related work) is very generic and could be revised (or omitted all-together) to focus more on SSL method sin CNNs (like par. 2). Similar, a part of the related work should be devoted to stochastic regularization techniques (pooling, dropout etc.) and data augmentation. - The network parameter description (line 155) can be confusing (especially if this is typeset in math mode). Is there a way to make the description more compact/descriptive? - Sec 4.2 and 4.3 could be merged (or shortened) as they describe the exact same experimental settings on two different datasets (SVHN and NORB). Comments on experiments: - Baselines (e.g. line 151/152): The comparisons involve training without augmentation (supervised) and with augmentation for the unsupervised loss (semi-supervised). There should be anther baseline in my opinion -- training on the entire augmented labeled set (supervised with data augmentation). This would be equivalent to adding a loss like \sum_j |y -f(T^j x)| for each x in the training set, thus letting the CNN encode the transformations in the weights. - Comparisons with Ladder networks [30] (Table 1): The comparison might not be exactly fair since the results are directly taken from Table 2 in [30] (i.e. not reproduced, for example, for the 100 sample training sets or with dropout/fractional max pooling etc.). In addition, the baseline CNN (supervised) for ladder net is different (supervised error is larger). Thus the absolute error comparison here might be misleading --a relative error decrease (with respect to the supervised network used as baseline for each model) might be better. Similar, the comparison in Table 4 is without data augmentation for the ladder networks. Secondary/minor comments and typos: - Many references have missing entries or only dates. Please revise. Update ImageNet ref [5]. - line 69: "Xavier's method"? - line 35: "more stable generalization"?: stability is generalization

Confidence in this Review

2-Confident (read it all; understood it all reasonably well)


Reviewer 2

Summary

In this work, the authors study semi-supervised learning with convolutional neural networks. They propose an unsupervised loss function that minimizes the difference between predictions of multiple passes of a training sample through a deep network. The new method is evaluated on some image recognition tasks and experimental results are shown.

Qualitative Assessment

Overall, the work is interesting and the authors do a nice job of setting up and reporting on multiple benchmark image recognition datasets. However, technical contributions of the paper could be summarized as adding a simple technique to reduce variation in predictions for the same training example due to techniques such as randomization, dropout, max-pooling, etc. A lot of the effort seems to have been spent on running experiments to demonstrate the effect of this extension to the loss function. One thing was unclear --- in practice, how did you optimize the new loss function? Do you replicate training samples in each mini-batch and keep copies of the predictions per sample from previous iterations in order to optimize the transformation function? What effect does this have on the speed and convergence of the network? Accuracies for the new method on the ImageNet ILSVR task seem significantly lower than current state-of-the-art methods.

Confidence in this Review

3-Expert (read the paper in detail, know the area, quite certain of my opinion)


Reviewer 3

Summary

The paper proposes and investigates a very simple idea for deep semi-supervised learning. The idea is to impose additional constraints derived from unlabelled data enforcing that deep representation of a data sample should be as constant as possible under nuisance transformations. The idea is thoroughly validated and is shown to achieve state-of-the-art on a number of datasets.

Qualitative Assessment

The idea is very simple and takes half a page (top half of page 3), while the bulk of the paper is devoted to its validation (which is the right thing to do). The validation brings spectacularly good results for semi-supervised learning across a range of standard datasets. The high absolute numbers are also explained by the use of sophisticated state-of-the-art architectures, however the authors are careful to show the advantage over the baseline that does not use the proposed augmentation or uses a competing idea (mutual-exclusivity) only. In terms of novelty, perhaps, the authors should have paid more attention to the link with [31], as their approach seems to be quite related (note that [31] have tried up to 32000 surrogate classes in their experiments, which is essentially very similar to what is proposed here). The approach is also very much related to tangent-distance methods [Simard et al.98]. Another obvious connection is "slow feature analysis" that has been investigated and tried with ConvNets many times. Finally, very similar trick has been used by [Kulkarni et al. NIPS15] (see section 3.2 there). Thus, the idea is not as novel as the submission tries to present. This, however, is compensated by the in-depth experimental validation and comparison.

Confidence in this Review

2-Confident (read it all; understood it all reasonably well)


Reviewer 4

Summary

The main idea behind this paper is that for any training sample, a good model should make the same prediction under any random transformation to the data or perturbation to the model. They propose an unsupervised loss function that they called the transformation/stability loss which explicitly tries to minimize the sum of squared differences between each pair of predictions resulting from different passes of the same sample through the model. They combine this with another unsupervised loss function, the mutual-exclusivity loss function, and show in several experiments that training in a semi-supervised fashion using these unsupervised losses in conjunction with a supervised loss function allows for better models to be trained when labeled data is not abundant. The datasets used to demonstrate this approach include MNIST, CIFAR10, CIFAR100, SVHN, NORB, and ILSVRC.

Qualitative Assessment

This paper presents an intuitive and straightforward idea about how to use unlabeled data when training deep convolutional networks. They demonstrate impressive results on benchmark datasets, including state-of-the-art performance on CIFAR10 and CIFAR100. However, their results on ImageNet lag far behind the current state-of-the-art, which makes me wonder if their approach can help in settings where training data is abundant, or if it is only relevant for specific settings where labeled data is limited. The paper is well-written and easy to follow for the most part. Here are some specific suggestions for improvements: - At the bottom of page 5 in Section 4.2 SVHN, it’s unclear to me what exactly was done in the experiment using sparse convolutional networks. It says “we create five sets of labeled data” and that “for each set, we randomly pick a different 1% subset of training samples.” Does this mean that the 1% is used as labeled training examples while the rest is used as unlabeled training examples? Are the results reported in Table 2 averaged over these five sets? - Why is data augmentation used sometimes and not others?

Confidence in this Review

2-Confident (read it all; understood it all reasonably well)


Reviewer 5

Summary

The authors present a semi-supervised procedure that uses the internal consistency of the data to drive the adaptation process of a learning machine. While traditional learning procedures only consider the difference between a required output and the actual output, this work also uses the fact that in classification problems the output should be the same despite the variations in the inputs and the learning machine. In other words, the machine's final goal is to find those invariances that maximize the recognition success.

Qualitative Assessment

The paper is very focused and well explained. The core idea makes a great deal of sense, and exploits information present in the data in ways that has been overlooked before. This allowed the authors to obtain impressive results with small labelled sets, and large unlabeled ones. This work is one step more towards understanding machine learning as finding invariances as pointed out by Gens and Domingos in their "Deep Symmetry Networks" paper. Minor comment: 1. Define the index i in eq. (1).

Confidence in this Review

3-Expert (read the paper in detail, know the area, quite certain of my opinion)